# Systematic Evaluation of Normalization Methods for Glycomics Data Based on Performance of Network Inference

**DOI:** 10.3390/metabo10070271

**Published:** 2020-07-02

**Authors:** Elisa Benedetti, Nathalie Gerstner, Maja Pučić-Baković, Toma Keser, Karli R. Reiding, L. Renee Ruhaak, Tamara Štambuk, Maurice H.J. Selman, Igor Rudan, Ozren Polašek, Caroline Hayward, Marian Beekman, Eline Slagboom, Manfred Wuhrer, Malcolm G. Dunlop, Gordan Lauc, Jan Krumsiek

**Affiliations:** 1Department of Physiology and Biophysics, Institute for Computational Biomedicine, Englander Institute for Precision Medicine, Weill Cornell Medicine, New York, NY 10022, USA; elb4003@med.cornell.edu; 2Institute of Computational Biology, Helmholtz Zentrum München—German Research Center for Environmental Health, 85764 Neuherberg, Germany; nathalie_gerstner@psych.mpg.de; 3Max Planck Institute for Psychiatry, 80804 Munich, Germany; 4Genos Glycoscience Research Laboratory, 10000 Zagreb, Croatia; mpucicbakovic@genos.hr (M.P.-B.); glauc@genos.hr (G.L.); 5Faculty of Pharmacy and Biochemistry, University of Zagreb, 10000 Zagreb, Croatia; tkeser@pharma.hr (T.K.); tpavic@unizg.hr (T.Š.); 6Biomolecular Mass Spectrometry and Proteomics, Bijvoet Center for Biomolecular Research and Utrecht Institute for Pharmaceutical Sciences, University of Utrecht, 3584 CH Utrecht, The Netherlands; k.r.reiding@uu.nl (K.R.R.); mauriceselman@yahoo.com (M.H.J.S.); 7Center for Proteomics and Metabolomics, Leiden University Medical Center, 2333 ZC Leiden, The Netherlands; lruhaak@ucdavis.edu (L.R.R.); m.wuhrer@lumc.nl (M.W.); 8Department of Clinical Chemistry and Laboratory Medicine, Leiden University Medical Center, 2333 ZC Leiden, The Netherlands; 9Usher Institute of Population Health Sciences and Informatics, University of Edinburgh, Edinburgh EH8 9AG, UK; irudan@hotmail.com; 10Medical School, University of Split, 21000 Split, Croatia; opolasek@gmail.com; 11Gen-Info Ltd., 10000 Zagreb, Croatia; 12Medical Research Council Human Genetics Unit, Institute of Genetics and Molecular Medicine, University of Edinburgh, Edinburgh EH4 2XU, UK; caroline.hayward@igmm.ed.ac.uk; 13Section of Molecular Epidemiology, Leiden University Medical Center, 2333 ZC Leiden, The Netherlands; m.beekman@lumc.nl (M.B.); p.slagboom@lumc.nl (E.S.); 14Colon Cancer Genetics Group, Institute of Genetics and Molecular Medicine, University of Edinburgh and Medical Research Council Human Genetics Unit, Edinburgh EH8 9YL, UK; alcolm.dunlop@igmm.ed.ac.uk

**Keywords:** glycomics, data normalization, gaussian graphical models

## Abstract

Glycomics measurements, like all other high-throughput technologies, are subject to technical variation due to fluctuations in the experimental conditions. The removal of this non-biological signal from the data is referred to as normalization. Contrary to other omics data types, a systematic evaluation of normalization options for glycomics data has not been published so far. In this paper, we assess the quality of different normalization strategies for glycomics data with an innovative approach. It has been shown previously that Gaussian Graphical Models (GGMs) inferred from glycomics data are able to identify enzymatic steps in the glycan synthesis pathways in a data-driven fashion. Based on this finding, here, we quantify the quality of a given normalization method according to how well a GGM inferred from the respective normalized data reconstructs known synthesis reactions in the glycosylation pathway. The method therefore exploits a biological measure of goodness. We analyzed 23 different normalization combinations applied to six large-scale glycomics cohorts across three experimental platforms: Liquid Chromatography-ElectroSpray Ionization-Mass Spectrometry (LC-ESI-MS), Ultra High Performance Liquid Chromatography with Fluorescence Detection (UHPLC-FLD), and Matrix Assisted Laser Desorption Ionization-Furier Transform Ion Cyclotron Resonance-Mass Spectrometry (MALDI-FTICR-MS). Based on our results, we recommend normalizing glycan data using the ‘Probabilistic Quotient’ method followed by log-transformation, irrespective of the measurement platform. This recommendation is further supported by an additional analysis, where we ranked normalization methods based on their statistical associations with age, a factor known to associate with glycomics measurements.

## 1. Introduction

Glycans have been recognized to contribute to the pathophysiology of every major disease [1]. To keep up with the increasing interest to better understand the involvement of glycans in biological processes at a molecular level, high-throughput platforms have been developed in the recent past. These platforms allow to profile glycans in large-scale datasets and from a wide variety of biospecimens. 

Similar to all other omics data types, glycomics samples need to be preprocessed prior to statistical analysis in order to minimize intrinsic, non-biological variation. This variation can arise, for example, from fluctuations in the instrument settings, sample preparation, or experimental conditions. The process that aims at reducing technical variations from the data is referred to as normalization. Different normalization procedures have substantially different assumptions regarding the nature of the non-biological variation, which, however, is unknown in most practical cases. Systematic comparisons of commonly implemented preprocessing strategies for various omics technologies have been published in recent years, including transcriptomics [2], proteomics [3], as well as metabolomics [4,5,6]. A recent study to guide the choice of normalization strategies for glycomics data has recently been published [7]; however, that study could not identify an optimal preprocessing strategy. Therefore, there is still no consensus on the appropriate normalization methods for glycomics data. 

This need for a glycomics-specific evaluation is further supported by the observation that the de facto standard for large-scale glycomics data preprocessing is Total Area (TA) normalization [8], which describes each glycan intensity in a sample as a percentage of the total. Following this transformation, the normalized intensities of a sample sum up to one (or 100%) by definition, leading to the loss of one degree of freedom. The division of each value by the sum of all values in a sample is referred to as a closure operation, and the resulting dataset is known as a compositional dataset [9]. Notably, these types of data normalization alter the structure of the covariance matrix, subsequently affecting any downstream correlation-based analysis (for details on this phenomenon, see the Methods Section). Compositional datasets are not unique to glycomics, but also widely occur in other fields, prominently in microbiome profiling [10], where percentages are used to describe the relative abundance of different microbial species. Notably, regular multivariate methods are not appropriate to treat these types of data, and specific statistical techniques need to be employed [11,12,13,14,15]. Most of such techniques require the definition of new variables, typically defined as ratios between the original compositional values [16,17,18]. This makes interpretation of the results in terms of the original quantities challenging [19,20]. 

In order to be able to infer biological interactions from the analysis of large-scale glycomics data, the selection of a more suitable alternative to TA normalization is therefore necessary. Given the variety of possible preprocessing strategies available, we need to define a criterion to quantitatively evaluate the performance of each method to select the most appropriate normalization method. 

Common evaluation schemes for the performance of preprocessing strategies are mostly based on two approaches: (1) Minimizing the variation between technical replicates [21,22], and (2) maximizing the variation across groups [6]. Consistency across technical replicates is a desirable outcome, but alone is not sufficient to guarantee good data quality, and technical replicates might not always be available. The maximization of variation across groups, on the other hand, is a viable measure that provides insights into the recovery of true biological signals. 

In this paper, we address the question of evaluating normalization strategies for glycomics data with a different, innovative approach. We assess the quality of a normalized dataset by its ability to reconstruct a biochemically correct pathway using statistical network inference. One popular approach for the inference of biological interactions is based on Gaussian Graphical Models (GGMs) [23]. GGMs depict correlating variables in the form of a network, where nodes represent the measurements (e.g., glycans) and edges represent their statistical associations. Specifically, GGMs quantify pairwise associations via partial correlations, an extension of regular Pearson correlations that accounts for the presence of confounding factors. Molecular measurements are generally highly correlated and thus contain a large number of correlations that are indirect and mediated by one or more other variables. Partial correlations allow to remove these indirect correlations automatically. Due to this property, GGMs have been repeatedly shown to selectively identify single enzymatic steps in metabolic [24,25] and glycosylation pathways [26], hence providing a reliable data-driven approach to infer biochemical pathways.

In this paper, we exploit the ability of GGMs to reconstruct biochemical reactions to define a biological measure of normalization quality. The idea is to compare the GGMs inferred from data normalized with different approaches to the known biochemical pathway of glycan synthesis and evaluate the quality of each normalization according to how well the corresponding GGM retrieves known synthesis reactions (Figure 1). By computing the overlap between estimated GGM and glycosylation pathway, we rely on a biological measure of quality, as a higher overlap indicates data whose correlations are able to better reflect known biochemical interactions. Hence, the normalization that produces the highest overlap is defined as the best. Glycomics data provide an ideal test case to demonstrate the validity of this approach, as the known biochemical pathway of synthesis is well characterized.

We compared the performance of different variations of seven commonly implemented normalization methods on data from six cohorts across three different glycomics platforms, including measurements of the Fragment crystallizable (Fc) region of Immunoglobulin G (IgG), total IgG, or total plasma N-glycans. In order to assess how our approach compares to other common normalization evaluation strategies, we additionally investigated how the normalization methods affect the statistical associations of glycans with age.

## 2. Results

### 2.1. Data

We analyzed six large-scale glycomics datasets (Table 1), measured on three different platforms:(1)In four cohorts (Korčula 2013, Korčula 2010, Split, Vis) [27], N-glycans from the Fc region of IgG were measured via liquid chromatography-electroSpray ionization-mass spectrometry (LC-ESI-MS). This platform allows to quantify glycopeptides, i.e., short amino acid sequences in proximity of the glycosylation site in combination with the attached glycans. Since IgG has four isoforms (also referred to as subclasses), which differ in their amino acid sequences [28,29], the LC-ESI-MS technology is able to distinguish among glycans bound to different IgG subclasses. In total, 50 N-glycopeptide structures were quantified: 20 for IgG1, 20 for IgG2 and IgG3 (which have the same glycopeptide composition and hence are not distinguishable by mass [28,29]), and 10 for IgG4. In the main manuscript, we show results for the Korčula 2013 cohort, which included 669 samples.(2)In one cohort (Study of Colorectal Cancer in Scotland; SOCCS) [30], IgG N-glycans were measured via ultra-high-performance liquid chromatography with fluorescence detection (UHPLC-FLD). In this case, all glycans bound to the IgG protein are first released and then measured, including the ones in the Fab region (see the Methods Section), but no information about the IgG subclass of origin is retained. Peaks in the chromatogram reflect chemical–physical properties of the measured molecules and not necessarily single glycan structures. In the specific case of IgG N-glycans, however, each UHPLC peak typically includes one highly predominant structure [31]. For the purpose of the analyses presented in this paper, we only considered the most abundant structure within each peak. The final UHPLC cohort consisted of 24 glycan peaks quantified in 535 samples.(3)In the last cohort (Leiden Longevity Study; LLS) [32], N-glycans from the whole set of human plasma proteins were measured via matrix-assisted laser desorption/ionization–Fourier-transform ion cyclotron resonance–mass spectrometry (MALDI-FTICR-MS). In this setting, glycans from all plasma proteins are released and measured together. Therefore, glycans originating from highly abundant and highly glycosylated proteins will be predominant. Notably, this platform only identifies molecular masses, so structural information is not directly available from the data. Therefore, within each mass, multiple glycan structures can be present, and this has to be taken into account. In the analyzed cohort, 61 distinct masses were quantified in 2056 samples.

### 2.2. Overview of Normalization Methods

Seven basic preprocessing approaches were considered, all of which are commonly used in omics data analysis (Table 2): (1) Raw (unprocessed) data were included for comparison, (2) quantile [33] and (3) rank [34] normalization are widely used in microarray data analysis, (4) Total Area (TA) is often used to preprocess large-scale glycomics [35] and microbiomics data [10], (5) median centering [4], and (6) Probabilistic Quotient normalization applied to raw and (7) to TA-normalized data are popular methods for the preprocessing of metabolomics data [36,37].

Since omics data have frequently been reported to follow an approximately log-normal distribution [38,39], and since GGMs assume normally distributed data, log-transformation of normalized data was also included in the analysis when applicable (indicated by a check mark in the second column of Table 2). This resulted in a total of 13 different preprocessing strategies. For LC-ESI-MS IgG data, 10 additional variations were included, as in this case, data normalization can be performed over the full dataset or per IgG subclass separately (third column in Table 2). A detailed description of each normalization procedure can be found in the Methods Section.

### 2.3. Prior Knowledge-Based Evaluation

Once all normalizations were applied to the data, partial correlation coefficients were computed with the GeneNet algorithm, which has been proven to give more reliable and stable estimates of partial correlation coefficients than the analytical method [40]. Statistical significance of coefficients was determined by applying a False Discovery Rate (FDR) of 0.01. The resulting partial correlation network, or Gaussian Graphical Model (GGM), was then compared to the respective biochemical pathway of glycan synthesis. As a quantitative measure of overlap between the calculated GGM and the pathway, we chose the Fisher test *p*-value (see the Methods Section), where lower *p*-values correspond to a higher overlap between inferred network and prior knowledge, thus corresponding to a better normalization. The biochemical pathway for IgG was taken from Benedetti et al. [26] (Figure 2), while the reference pathway for the total-plasma N-glycome was based on the measured glycan masses (see the Methods Section). Schematics of the pathways used for the evaluation can be found in Appendix A.

### 2.4. LC-ESI-MS—IgG Fc N-Glycopeptides

For the LC-ESI-MS platform, most methods produced networks with high overlap to the biochemical pathway of synthesis, indicated by low Fisher’s test *p*-values (Figure 3 and Appendix A, left).

Interestingly, the unprocessed data (Raw) were among the best-performing methods, which might be related to the fact that, with this platform, the ionization is dominated by the peptide, which might serve as an internal standard for the glycan quantification. As expected, TA-based normalizations performed significantly worse than all other considered strategies, probably due to the alteration of the covariance matrix induced by closure operation. Moreover, we observed that in most cases, log-transformation did not improve performance (Figure 3 and Appendix A, center), with the exception of the Vis cohort, where log-transformed data seemed to perform marginally better overall (Appendix A, center). Given the assumption of normality of the Gaussian graphical models, we expected log-transformed data, which are more normally distributed, to perform better than their non-transformed counterparts. This might indicate that GGMs, although formally only suitable for normally distributed data, are also effective for non-Gaussian data. An exception to this observation was the TA-log normalization, for which log-transformation appears to neutralize the constraints imposed by TA and hence, improves performance. Normalizing per total IgG or per IgG subclass did not result in substantial differences in performance, except for TA (Figure 3, right).

In summary, we showed that for LC-ESI-MS IgG Fc glycomics data, all considered preprocessing performed comparably except TA, which was significantly worse than the rest. Moreover, non-log-transformed data did not perform worse than the transformed data, and normalizing per total IgG or per IgG subclass did not make a significant difference.

### 2.5. UHPLC-FLD—Total IgG N-Glycans

For the UHPLC-FLD dataset, contrary to the previous case, the performance was highly affected by the chosen normalization method (Figure 4, left), with TA Probabilistic Quotient and Probabilistic Quotient ranking at the top. In this case, the unprocessed data performed poorly. Moreover, in contrast to what we observed in the LC-ESI-MS case, for UHPLC-FLD data, the log-transformation had a significant impact on the performance of normalizations, although with opposite effects depending on the methodology: for some it substantially enhanced performance (Quantile, Total Area), while for others it was detrimental (Rank, Raw data) (Figure 4, right).

### 2.6. MALDI-FTICR-MS—Total Plasma N-Glycans

The MALDI dataset included 61 glycan peaks. Similar to the LC-ESI-MS case, most methods performed comparably (Figure 5, left). Log-transformed unprocessed data yielded the worst performance, although in all other cases, log-transformation did not significantly affect the normalization performance (Figure 5, right).

In conclusion, for MALDI data, most normalization methods performed comparably. Log-transformation did not significantly alter performance, except when considering log-transformed unprocessed data, which was the worst performing approach.

### 2.7. Comparison with Phenotype Association Analysis

As a final step, we investigated how our method relates to a more common normalization evaluation strategy: The maximization of the statistical association between normalized values and a given phenotype. We chose age as the phenotype for association, since age has been repeatedly shown to strongly correlate with glycan abundances in blood [41,42,43,44,45].

For each of the six analyzed datasets and for each normalization strategy, we computed the *p*-values of the statistical associations between glycan abundances and age and subsequently corrected for multiple testing (Appendix A, Appendix A). To summarize the results, we calculated the fraction of significant associations (False Discovery Rate 0.01) per normalization strategy and averaged across datasets (Table 3). Similar to our network-based analysis, most normalization performed comparably in the LC-ESI-MS datasets, displaying a large number of significant associations. In contrast, in the other two datasets, the fraction of significant associations was heavily dependent on the normalization method. On average, the preprocessing strategies based on Probabilistic Quotient produced the highest number of significant associations, substantially outperforming all other approaches. These findings are consistent with our network-based approach, corroborating our results.

## 3. Discussion

Several systematic evaluations of preprocessing methodologies have been recently published for different omics data types, but glycomics has received little attention so far in this regard. In order to address this gap, we developed an innovative approach to assess the quality of different normalization strategies applied to glycomics data. The main feature of our procedure lies in the definition of a biological measure of quality. More specifically, we quantify how well significant correlations in the data normalized with a given technique represent known biochemical reactions in the pathway of glycan synthesis. Our quantitative measure of choice for this evaluation was the *p*-value of a Fisher’s exact test, which allows for an intuitive interpretation of overlap between correlations and biochemical pathway.

We performed a systematic analysis of 23 preprocessing strategies applied to six large-scale glycomics cohorts across three platforms, with measurements ranging from a single protein and single glycosylation site (LC-ESI-MS), to total plasma N-glycome (MALDI-FTICR-MS). The observed normalization ranking was consistent across platforms; overall, the Probabilistic Quotient appeared to be the most reliable method, as all variations of this procedure ranked consistently in the top performers in all cohorts and across platforms. Log-transformation and normalization per IgG subclass or per total IgG did not seem to significantly affect the ability of this method to correctly retrieve the glycan synthesis pathway. Interestingly, while Total Area normalization did not rank high in comparison to other methods (as expected), the log-transformed Total Area preprocessing was a well-performing method. In fact, TA Probabilistic Quotient was among the best performing approaches overall, suggesting that additional transformations on TA normalized data can neutralize the constraints imposed on the data correlation structure, as shown in Dieterle et al. [36].

One interesting finding was the substantial difference of the evaluation results between MS- and UHPLC-based platforms: While for MS, most normalization approaches performed comparably, the variance among the considered strategies was considerable for UHPLC. The origin of this discrepancy is not easy to trace, but it could be due to the fact that UHPLC does not separate glycans according to their mass, like MS-based techniques do, but according to their chemical and physical properties. This leads most chromatographic peaks to represent a mixture of glycan structures. Although it has been shown that there is a predominant structure in the vast majority of IgG chromatographic peaks [31], this contamination is likely to make the data correlation structure noisier and thus more sensitive to different normalizations. Moreover, it is expected to affect the comparison to the biological reference, which does not account for any structure mixture.

While our results seem to suggest that log-transformation does not significantly affect performance, it should be considered that data normality is an assumption for many other statistical tests and approaches, and thus we still recommend to always log-transform omics data after normalization.

To assess how our approach compares to a more common normalization evaluation strategy, we ranked the preprocessing methods based on how strongly the normalized abundances associated with age. Consistent with our network-based results, Probabilistic Quotient-based approaches clearly outperformed all other methods.

The network approach described in this paper could be employed to evaluate normalization strategies in other types of mass-flow data, e.g., metabolomics data. Moreover, we could extend this approach to evaluate other preprocessing steps. For example, it has already been shown that, for untargeted metabolomics data, different missing value imputation strategies have a prominent impact on the results of the downstream analysis [46]. We could investigate whether the same holds for glycomics data and quantitatively evaluate the performance of each strategy. Similarly, our framework could be applied to the evaluation of batch correction approaches, which aim at reducing the technical variation due to samples being measured at different times.

In conclusion, we recommend normalizing glycan data with the Probabilistic Quotient normalization followed by log-transformation. This technique was robust and reliable regardless of the measurement platform.

## 4. Materials and Methods

### 4.1. Datasets

#### 4.1.1. LC-ESI-MS

Samples were collected from the Croatian islands of Vis and Korčula, and were obtained from the “10,001 Dalmatians” biobank [27], while samples for a second cohort from Korčula and a cohort from Split were collected separately a few years later. For this paper, we only considered unrelated individuals, as described previously [26]. Samples with missing values were excluded from this analysis. The final datasets included 669 (Korčula2013), 504 (Korčula 2010), 980 (Split), and 395 (Vis) samples. The Croatian cohorts received ethical approval of the ethics committee of the University of Split School of Medicine, as well as the South East Scotland Research. Written informed consent was obtained from each participant. A detailed description of the experimental procedure can be found in Selman et al. [47] and Huffman et al. [48].

#### 4.1.2. UHPLC-FLD

The Study of Colorectal Cancer in Scotland (SOCCS) (1999–2006) is a case–control study designed to identify genetic and environmental factors associated with nonhereditary colorectal cancer risk and survival outcomes [49]. Only the control samples with no missing values were considered for this analysis, for a total of 535 samples. Approval for the study was obtained from the MultiCentre Research Ethics Committee for Scotland and Local Research Ethics committee, and all participants gave written informed consent. A detailed description of the experimental procedure can be found in Vučković et al. [30].

#### 4.1.3. MALDI-FTICR-MS

The Leiden Longevity Study (LLS) is a family-based study comprising 1671 offspring of 421 nonagenarians sibling pairs of Dutch descent, and the 744 partners of these offspring [50]. After removal of samples with missing values, a total of 2056 individuals were included in the current analysis. The study protocol was approved by the Leiden University Medical Center ethical committee and an informed consent was signed by all participants prior to participation in the study. A detailed description of the experimental procedure can be found in Reiding et al. [32].

### 4.2. Normalization Methods

Prior to normalization, samples containing missing values were excluded from all cohorts.

Raw: These are the unprocessed, raw peak intensities.

Median Centering: The median value over all samples is subtracted from each glycan value in the dataset. The underlying assumption is that the samples have a constant offset.

Total Area: The intensity of each glycan is normalized to the total area of the spectrum. This preserves the relative intensities of each peak within the sample, at the cost of losing one degree of freedom due to the constant sum constraint and giving rise to a so-called “compositional dataset” [51]. The underlying assumption here is that only relative intensities are biologically relevant. This transformation, however, introduces artifacts in the covariance matrix, which, just because of the constraint introduced by the normalization, results with at least one negative value per each row [9].

Probabilistic Quotient: This approach is based on the calculation of the dilution factor of each sample with respect to a reference sample [36]. Here, the reference sample was calculated as the median value of each glycan’s abundance across all measured samples. For each sample, a vector of quotients was then obtained by dividing each glycan measure by the corresponding value in the reference sample. The median of these quotients was then used as the sample’s dilution factor, and the original sample values were subsequently divided by that value. The underlying assumption is that the different intensities observed across individuals are imputable to different amounts of the biological material in the collected samples.

Quantile: This method forces the distributions of the glycans (columns) to be the same with respect to the quantiles [52]. It requires replacing each point of a glycan with the mean of the corresponding quantile, resulting in perfectly aligned distributions by construction.

Rank: Values are replaced with their corresponding ranks across the samples.

Log-transformation: Biological data have been observed to often follow a log-normal distribution [38]. Since our correlation estimator assumes normally distributed data, we included both the non-transformed and the log-transformed data for each considered normalization method, except the median centering.

Subclass-specific normalization: LC-ESI-MS measures IgG glycosylation at the glycopeptide level, which means that the information about the IgG isoform is preserved. In Caucasian populations, as those considered in this paper, the Fc glycopeptides of IgG2 and IgG3 have identical peptide moieties [28,29], and are therefore not distinguishable with this profiling method. Furthermore, only 10 glycoforms of IgG4 were detectable due to the low abundance of this IgG subclass in human plasma. For this platform, each normalization method was applied both on the 50 glycoform measurements together, as well as separately per each IgG subclass.

### 4.3. Prior Knowledge

The IgG N-glycan synthesis pathway considered in this analysis reflects the extended version established and validated in Benedetti et al. [26]. For LC-ESI-MS data, the same glycosylation pathway was assumed for all IgG subclasses (Figure 2). For UHPLC-FLD data, each peak was approximated to only be represented by its most abundant structure, according to Pučić et al. [31] (Appendix A).

For MALDI-FTICR-MS, the biochemical pathway was constructed based on current understanding of glycosylation synthesis reactions [53] (Appendix A). However, since the available data included only glycan masses and not single structures, all the structures with the same mass were merged into a single node and masses not included in our dataset were removed (Appendix A). The resulting compositional pathway was then adapted to match the masses in the dataset (Appendix A).

### 4.4. GGM Estimation

Correlation networks were computed using the preprocessed glycan abundances. GGMs are based on partial correlation coefficients, which represent pairwise dependencies in multivariate normally distributed data when conditioned against all other variables. To obtain a reliable estimate for the partial correlation matrix, we used the shrinkage-based GeneNet algorithm [40]. Multiple hypothesis testing was corrected for by controlling the FDR at 0.01 using the Benjamini–Hochberg method [54].

### 4.5. Overlap to the Biological Reference

The overlap between biological reference and correlation network was calculated using Fisher’s exact tests [55,56], which evaluate whether two categorical variables are statistically independent [57], with low *p*-values indicating a lack of independence. We classified all glycan pairs in a 2 × 2 contingency table, according to whether they were connected by an edge in both the data-driven GGM and the biochemical pathway (true positives), only in the GGM (false positives), only in the pathway (false negatives), or in neither (true negatives). From these values, the computed Fisher’s exact test *p*-value can be interpreted as an overlap measure between the two classifiers (in our case, represented by the presence or absence of an edge in the GGM and in the pathway). The lower the *p*-value, the higher the overlap. In the context of this paper, the normalization with the lowest Fisher’s test *p*-value will produce the GGM with the highest overlap to the biochemical pathway of glycan synthesis and will be ranked as the best normalization.

### 4.6. Statistical Association with Age

For each platform and normalization method, we computed the *p*-value of the linear model glycan ~ age. *p*-values were corrected for multiple testing by controlling the false discovery rate (FDR) at 0.01. For the purpose of this analysis, we only considered the normalization approaches common to all datasets for easier comparison. We then computed the fraction of significant *p*-values for each dataset and normalization method. Since we analyzed four LC-ESI-MS cohorts versus one UHPLC-FLD and one MALDI-FTICR, we first computed the average of the fractions of significant associations across the LC-ESI-MS cohorts and then averaged those values with the other two cohorts. This final average is reported in Table 3.

## Figures and Tables

**Figure 1 metabolites-10-00271-f001:**
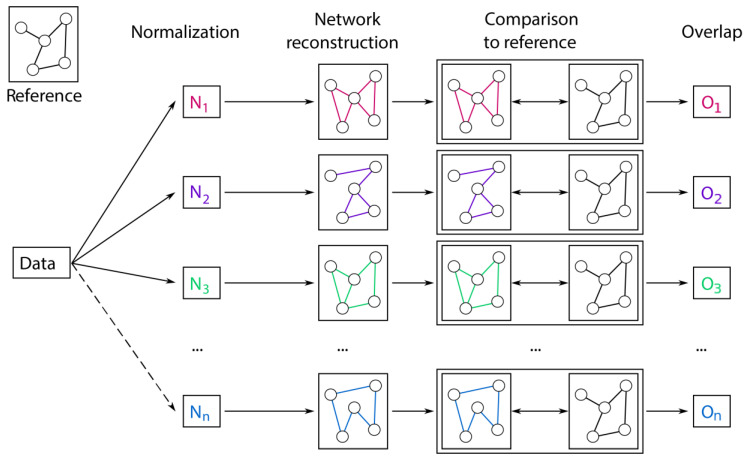
Pipeline for the evaluation of different normalization methods for glycomics data. First, data are normalized with various approaches. From each processed dataset, a Gaussian Graphical Model (GGM) is inferred and compared to the known biochemical pathway of glycan synthesis. The result of this comparison is a quantitative overlap value that describes how well the estimated GGM represents known synthesis reactions. This overlap is then used to evaluate the normalization approach, where higher overlap corresponds to a better data normalization.

**Figure 2 metabolites-10-00271-f002:**
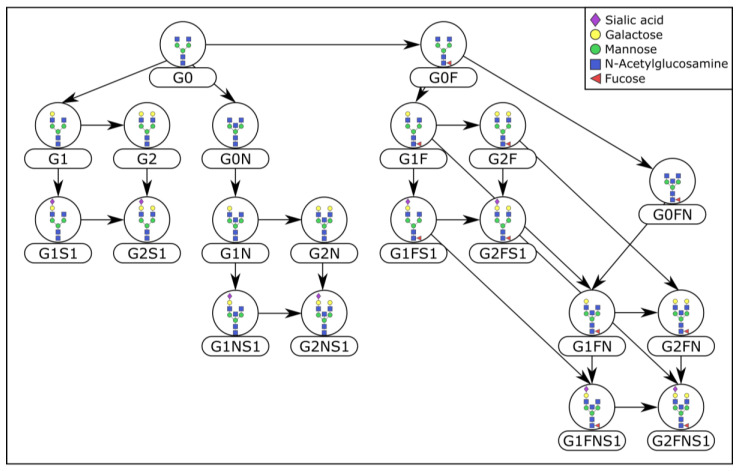
Reference pathway for Immunoglobulin G (IgG) Liquid Chromatography-ElectroSpray Ionization-Mass Spectrometry (LC-ESI-MS) data. IgG glycans include monosaccharides such as mannose, N-acetylglucosamine, galactose, fucose, and sialic acid, and are synthesized by the incremental addition of single monosaccharides.

**Figure 3 metabolites-10-00271-f003:**
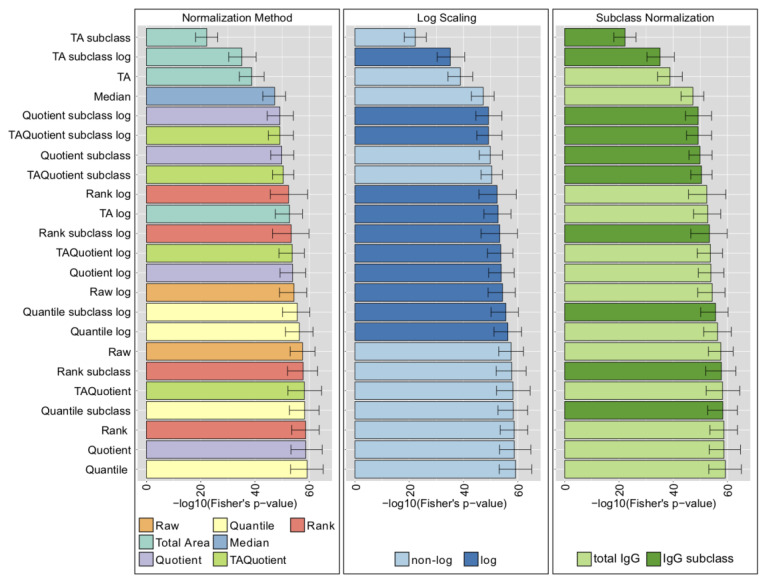
LC-ESI-MS normalization analysis results (Korčula 2013 cohort). Results in the panels are colored according to type of normalization (left), log-transformation (center), or normalization per IgG subclass or total IgG (right). Bars represent the median of the Fisher’s exact test *p*-values over 1000 bootstrap samples, and error bars indicate the corresponding 95% confidence intervals.

**Figure 4 metabolites-10-00271-f004:**
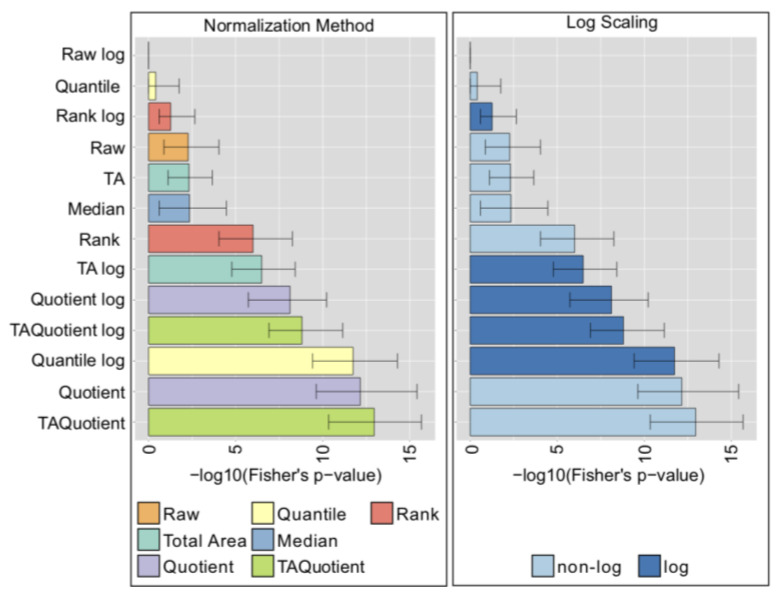
Ultra High Performance Liquid Chromatography with Fluorescence Detection (UHPLC-FLD) normalization analysis results (Colorectal cancer controls cohort). Results in the panels are colored according to type of normalization (left), or log-transformation (right). Bars represent the median of the Fisher’s exact test *p*-values over 1000 bootstrapping, and error bars indicate the corresponding 95% confidence intervals.

**Figure 5 metabolites-10-00271-f005:**
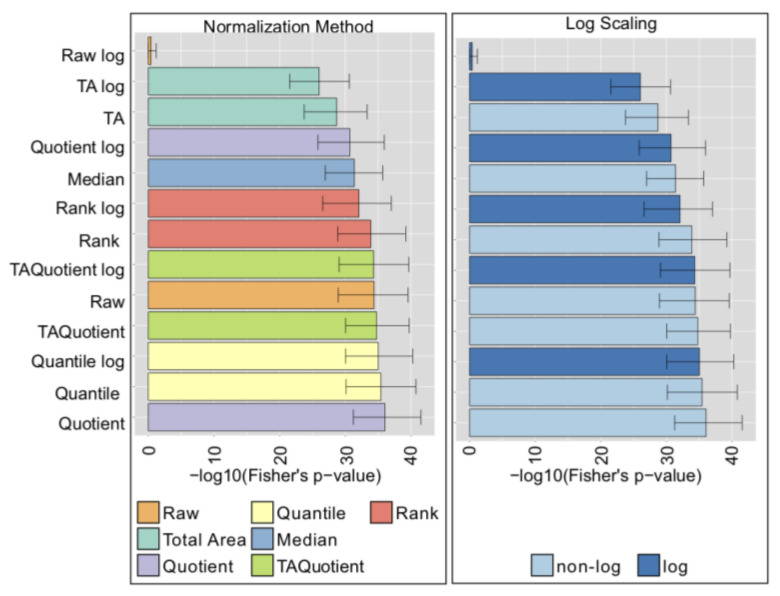
Matrix Assisted Laser Desorption Ionization-Furier Transform Ion Cyclotron Resonance (MALDI-FTICR-MS) normalization analysis results (Leiden Longevity Study cohort). Results in the panels are colored according to type of normalization (left), or log-transformation (right). Bars represent the median of the Fisher’s exact test *p*-values over 1000 bootstrapping, and error bars indicate the corresponding 95% confidence intervals.

**Table 1 metabolites-10-00271-t001:** Summary of datasets.

	LC-ESI-MS				UHPLC-FLD	MALDI-FTICR-MS
Dataset Name	Korčula 2013	Korčula 2010	Split	Vis	CRC Controls	LLS
Glycans measured	IgG Fc	IgG Fc	IgG Fc	IgG Fc	IgG total	Total plasma
Number of peaks	50	50	50	50	24	61
Number of samples for analysis	669	504	980	395	535	2056
Age range (mean ± SD)	18–88 (53.2 ± 16.3)	18–98 (56.4 ± 13.6)	18–85 (50.3 ± 14.3)	18–91 (55.8 ± 15.2)	21–74 (51.6 ± 5.9)	30–80 (59.2 ± 6.7)

LC-ESI-MS: Liquid Chromatography-ElectroSpray Ionization-Mass Spectrometry; UHPLC-FLD: Ultra High Performance Liquid Chromatography with Fluorescence Detection; MALDI-FTICR-MS: Matrix Assisted Laser Desorption Ionization-Fourier Transform Ion Cyclotron Resonance-Mass Spectrometry; CRC: Colorectal cancer; LLS: Leiden Longevity Study; IgG: Immunoglobulin G; Fc: Fragment crystallizable; SD: standard deviation.

**Table 2 metabolites-10-00271-t002:** Evaluated normalization methods.

Normalization	Label	Group
Raw	Raw	Basic Normalizations
Quantile per glycan	Quantile	
Rank per glycan	Rank	
Total Area	TA	
Median Centering	Median	
Probabilistic Quotient	Quotient	
Total Area + Probabilistic Quotient	TAQuotient	
log(Raw)	Raw log	Logarithm
log(Quantile per glycan)	Quantile log	
log(Rank per glycan)	Rank log	
log(Total Area)	TA log	
log(Probabilistic Quotient)	Quotient log	
log(Total Area + Probabilistic Quotient)	TAQuotient log	
(Quantile per glycan) per IgG subclass	Quantile subclass	Per Subclass
(Rank per glycan) per IgG subclass	Rank subclass	
(Total Area) per IgG subclass	TA subclass	
(Probabilistic Quotient) per IgG subclass	Quotient subclass	
(Total Area + Probabilistic Quotient) per IgG subclass	TAQuotient subclass	
(log(Quantile per glycan)) per IgG subclass	Quantile log subclass	
(log(Rank per glycan) per IgG subclass	Rank log subclass	
(log(Total Area)) per IgG subclass	TA log subclass	
(log(Probabilistic Quotient)) per IgG subclass	Quotient log subclass	
(log(Total Area + Probabilistic Quotient)) per IgG subclass	TAQuotient log subclass	

IgG: Immunoglobulin G; log: logarithm.

**Table 3 metabolites-10-00271-t003:** Fraction of glycans significantly associated with age (False Discovery Rate 0.01). Normalization approaches are sorted by decreasing average fraction of significant associations.

Platform		LC-ESI-MS					UHPLC-FLD	MALDI-FTICR	
	Dataset	Korčula 2013	Korčula 2010	Split	Vis	LC-ESI-MS Average	CRC Controls	LLS	Weighted Average across Platforms
Normalization									
TAQuotient log		0.340	0.680	0.700	0.740	0.615	0.625	0.590	0.610
Quotient log		0.340	0.660	0.700	0.740	0.610	0.625	0.590	0.608
Quotient		0.320	0.660	0.740	0.680	0.600	0.583	0.574	0.586
TAQuotient		0.320	0.660	0.740	0.660	0.595	0.583	0.574	0.584
TA log		0.360	0.700	0.760	0.700	0.630	0.542	0.475	0.549
TA		0.300	0.720	0.780	0.720	0.630	0.500	0.475	0.535
Quantile		0.220	0.600	0.700	0.640	0.540	0.000	0.279	0.273
Raw		0.180	0.560	0.700	0.620	0.515	0.000	0.279	0.265
Rank		0.220	0.520	0.700	0.580	0.505	0.000	0.262	0.256
Quantile log		0.220	0.560	0.700	0.580	0.515	0.000	0.246	0.254
Median		0.220	0.520	0.560	0.640	0.485	0.000	0.246	0.244
Raw log		0.220	0.560	0.680	0.540	0.500	0.000	0.213	0.238
Rank log		0.140	0.400	0.620	0.540	0.425	0.000	0.115	0.180

## Data Availability

Unprocessed LC-ESI-MS and UHPLC-FLD glycomics data are publicly available from the figshare database with doi: 10.6084/m9.figshare.12581735. MALDI-FTICR-MS data can be accessed by signing a data transfer agreement. For details, please contact P. Eline Slagboom (P.Slagboom@lumc.nl) or Marian Beekman (m.beekman@lumc.nl). R code to replicate the findings reported in this paper is available from GitHub at https://github.com/krumsieklab/GlycoNorm.

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
