# Peer review of "Systematic Evaluation of Normalization Methods for Glycomics Data Based on Performance of Network Inference"

_metabolites, 2020, doi:10.3390/metabo10070271_

Round 1
Reviewer 1 Report
There are a lot of positives about this paper. The field could benefit from a comprehensive study of normalization methods. The authors’ approach is novel and interesting. The authors did an excellent job of being comprehensive in collecting data sets and comparing normalization methods: They picked the three most important data types to study. They compared numerous normalization methods. The paper has just one major flaw and two minor ones. If the authors can add data to address the major flaw, and validate that their method really identifies the best normalization strategy, the manuscript should be published.
Major Flaw:
The authors are using an untested approach for comparing the normalization methods. They get results from the approach, but they don’t prove that those results are correct. At a minimum, they should validate these findings by selecting an example data set and demonstrating either that the technical replicates are more similar when using the best normalization vs. the other options, or that a biological difference is easier to detect with the best normalization. They could pull a data set from the literature for this validation. Without doing this experiment, there is no validation that this untested method (of using the likely biological pathways for glycans to compare to an inferred GGM) actually produces a correct answer about normalization quality.
Minor points:
- The readers of this manuscript will mostly be analysts who study glycans. As a group, they will not be very familiar with GGMs; so they will not understand the approach used, based on the terse explanation in the introduction. The authors should include more detail (at least another paragraph) describing GGMs and what biological pathways they plan to map. It would help to move at least one of the pathways in the supplemental data into the main text. That will help people follow the experiment. Ex: the LC-ESI-MS pathway could be included.
- The conclusions are partially inconsistent with the data. On line 184, they said “the results… were consistent across all four cohorts”. But this is clearly not the case. In three of the four data sets, the log normalizations performed pretty badly, but in the Vis cohort, the six best methods were log-based normalizations. This discrepancy should be addressed.
Author Response
Response to Reviewer 1 Comments
Point 1: There are a lot of positives about this paper. The field could benefit from a comprehensive study of normalization methods. The authors’ approach is novel and interesting. The authors did an excellent job of being comprehensive in collecting data sets and comparing normalization methods: They picked the three most important data types to study. They compared numerous normalization methods. The paper has just one major flaw and two minor ones. If the authors can add data to address the major flaw, and validate that their method really identifies the best normalization strategy, the manuscript should be published.
Response 1: We have expanded the manuscript with a new result section to address the validation concern, as well as to clarify the minor points raised by the reviewer (see below for details). We have also marginally edited other sections of the manuscript (e.g., abstract, introduction, discussion and methods) to integrate the new additions in the argumentation. All changes have been tracked in the manuscript to facilitate review. We will make the glycan data and R code publicly available to the reader in case of acceptance.
Point 2: Major Flaw: The authors are using an untested approach for comparing the normalization methods. They get results from the approach, but they don’t prove that those results are correct. At a minimum, they should validate these findings by selecting an example data set and demonstrating either that the technical replicates are more similar when using the best normalization vs. the other options, or that a biological difference is easier to detect with the best normalization. They could pull a data set from the literature for this validation. Without doing this experiment, there is no validation that this untested method (of using the likely biological pathways for glycans to compare to an inferred GGM) actually produces a correct answer about normalization quality.
Response 2: We appreciate the point raised by the reviewer. We agree that clarifying how our evaluation relates to more commonly used normalization quality measures would be of great interest for the reader and would substantially improve the manuscript. One of the most popular normalization evaluation strategies is based on the maximization of the association between the normalized abundances and a given phenotype. Here, we chose to analyze age as our phenotype of choice. This was motivated by the observation that age has been repeatedly shown to strongly associate with protein N-glycans in blood 1–5.
For each of the six analyzed datasets and for each normalization strategy, we computed the associations between glycan abundances and age. After correcting the obtained p-values for multiple testing, we calculated the fraction of significant associations per normalization strategy and averaged across datasets. Our results show that the quotient normalization-based strategies outperform the other approaches, corroborating our network analysis-based results.
The results of this analysis are now reported in a new result section in the main text of the manuscript (section “2.7 Comparison with phenotype association analysis”).
Point 3: Minor points: 1. The readers of this manuscript will mostly be analysts who study glycans. As a group, they will not be very familiar with GGMs; so they will not understand the approach used, based on the terse explanation in the introduction. The authors should include more detail (at least another paragraph) describing GGMs and what biological pathways they plan to map. It would help to move at least one of the pathways in the supplemental data into the main text. That will help people follow the experiment. Ex: the LC-ESI-MS pathway could be included.
Response 3: Following the reviewer’s suggestion, we have expanded the description of GGMs in the introduction in order to make the idea behind our approach and the role of GGMs clearer to the reader. Moreover, we have now included the IgG glycosylation pathway used for the LC-ESI-MS data analysis in the main manuscript.
Point 4: 2. The conclusions are partially inconsistent with the data. On line 184, they said “the results… were consistent across all four cohorts”. But this is clearly not the case. In three of the four data sets, the log normalizations performed pretty badly, but in the Vis cohort, the six best methods were log-based normalizations. This discrepancy should be addressed.
Response 4: We apologize for this inconsistency in the argumentation. We have now edited our claims with more accurate statements about the results.
References
- Yu, X. et al. Profiling IgG N-glycans as potential biomarker of chronological and biological ages: A community-based study in a Han Chinese population. Medicine (Baltimore). 95, (2016).
- Krištić, J. et al. Glycans Are a Novel Biomarker of Chronological and Biological Ages. Journals Gerontol. Ser. A 69, 779–789 (2014).
- Šunderić, M. et al. Changes Due to Ageing in the Glycan Structure of Alpha-2-Macroglobulin and Its Reactivity with Ligands. Protein J. 38, 23–29 (2019).
- Ruhaak, L. R. et al. Plasma protein N-glycan profiles are associated with calendar age, familial longevity and health. J. Proteome Res. 10, 1667–1674 (2011).
- Vanhooren, V. et al. N-Glycomic Changes in Serum Proteins During Human Aging. Rejuvenation Res. 10, 521--531a (2007).
Reviewer 2 Report
The study by Benedetti et al is a timely assessment of different data normalization methods applied to existing large scale data sets of glycomic data obtained by mass spectrometry and LC analyses. It is novel on many levels, as any assessment like this has not been previously reported. The large sample numbers evaluated, as well as the multiple analytical platforms used, well represent the major strategies being used in developing glycomics for clinical utility. It is well written and concise for the scale that is covered, figures are clear, and conclusions are direct. There are no concerns or suggestions.
Author Response
Response to Reviewer 2 Comments
Point 1: The study by Benedetti et al is a timely assessment of different data normalization methods applied to existing large scale data sets of glycomic data obtained by mass spectrometry and LC analyses. It is novel on many levels, as any assessment like this has not been previously reported. The large sample numbers evaluated, as well as the multiple analytical platforms used, well represent the major strategies being used in developing glycomics for clinical utility. It is well written and concise for the scale that is covered, figures are clear, and conclusions are direct. There are no concerns or suggestions.
Response 1: We would like to thank the reviewer for the consideration.
Round 2
Reviewer 1 Report
The authors did an outstanding job of revising and improving the paper. It is ready for publication and will be a strong contribution to the journal. Thanks for doing such a nice job on the revisions.